# Reflectance of Oil Paintings: Influence of Paint Layer Thickness and Binder Amount

José Santiago Pozo-Antonio [1,*], Carolina Cardell [2], Sonia Sánchez [2] and Jesús Montes Rueda [2]

[1]  CINTECX, Grupo GESSMin, Departamento de Enxeñaría dos Recursos Naturais e Medio Ambiente, University of Vigo, 36310 Vigo, Spain
[2]  Department of Mineralogy and Petrology, Faculty of Science, University of Granada, 18071 Granada, Spain; cardell@ugr.es (C.C.); ssonia@ugr.es (S.S.); jesusmr@ugr.es (J.M.R.)
[*]  Correspondence: ipozo@uvigo.es

**Abstract:** Roughness, pigment impurities, and neoformed minerals are important factors affecting the reflectance of painted surfaces. However, other factors that have not yet been studied in detail, such as the total thickness of the paint layer and the amount of binder, should also be considered. In this research, oil painting mock-ups, each made with a different number of layers of paint containing a single pigment (lead white, orpiment, cinnabar, azurite or malachite) mixed with different proportions of oil binder, were examined using a hyperspectral imaging system. The results indicate that these characteristics do not directly influence the reflectance of the painted surface. Interestingly, we found that the distribution of the pigments and the oil binder in the paint system (and pigment-binder interaction) are also key to understanding the reflectance values. Thus, accumulation of oil on top of the uppermost paint layer in the multilayer painting mock-ups caused a reduction in reflectance. The increased translucency of the oil paint containing lead white pigment also modified the reflectance, possibly due to the formation of metal soap. Other factors found to affect the reflectance of the painting mock-ups are discussed.

**Keywords:** spectral imaging; oil painting; pigment; binder amount; painting thickness; pictorial heritage; non-invasive characterization; mock-up



## 1. Introduction

Reflectance spectroscopy is an appealing technique in the field of conservation of pictorial heritage and can be applied to real paintings and painting mock-ups. Non-invasive spectral imaging devices (such as multi- and hyper-spectral systems) can be used to identify the composition of materials (pigments and binders) and detect the presence of decay products and underlying preparatory drawings [1–19]. The influence of intrinsic properties of the pigments (composition and grain size) and of the binder (composition and proportion) on the reflectance values and spectra have been reported in the relevant literature [9–15]. In this regard, as stated in [15], when real paintings are considered, effects associated with pigment grain size, binder quantity or binder aging must be taken into account in order to understand the absorption and scattering that occurs on the surface of paintings.

In previous scientific publications concerning reflectance measurements applied to pictorial heritage, different researchers working with tempera (rabbit glue, egg and casein) or oil (linseed oil) painting mock-ups have reported the effects of pigment size and binder composition on the reflectance, either in terms of intensity or the shape of the reflectance spectra [9–14,16]. Different findings have been reported regarding the influence of the pigment size on the reflectance spectra. Thus, for tempera paints containing some pigments, e.g., azurite, the particle size of the pigment only affected the reflectance value (intensity), i.e., the finer the pigment the greater the reflectance value of the paint [9]. By contrast, for other pigments, such as malachite, the shape of the spectrum also showed small peak

shifts of up to 20 nm [9]. Other researchers [16] have reported that for casein-based paints, the intensity of reflectance increases with increasing pigment size, with small peak shifts for malachite and azurite pellets (i.e., powdered pigment prepared in pellet form without binder) with different pigment sizes. However, no shifts were detected for pellets of red and yellow pigments. The intensity of reflectance increased as the particle size in pigment pellets decreased, unlike in paint systems (different mixtures of binder/s and pigment/s) [16]. This different behavior was attributed to the presence of a binder (casein) with a much higher refractive index than that of air, as previously reported [10,13]. In a recent publication, these changes in reflectance of paints with different pigment sizes were related to the wide size range, pigment impurities (and the amount in the paint system) and/or neoformation of minerals during paint preparation [14]. Moreover, the importance of conducting detailed studies of the pigment by determining the main and secondary (when present) maximum particle sizes was emphasized. Likewise, evaluation of the pigment morphology is recommended in order to explain the physical properties (specifically color, roughness and reflectance) of paint systems [14]. Considering the binder, linseed oil (unlike egg yolk or rabbit glue) seems to increase paint saturation and induces less light scattering rather than increasing absorbance [17].

The present study specifically focused on the influence of the total thickness of the paint coating and of the amount of binder on the reflectance values, as determined by hyperspectral imaging, in order to identify the factors affecting the reflectance of the painted surfaces. With this aim, oil painting mock-ups containing linseed oil as binder and a single pigment (one of five pigments of different color, composition and grain size) were analyzed. Initially the powdered pigments were characterized in terms of particle size and mineralogical composition, determined by laser diffraction analysis and X-ray diffraction analysis, respectively. Prepared mock-up paintings were subsequently analyzed by stereomicroscopy and hyperspectral imaging; the reflectance spectra were studied in relation to the properties of the paint surfaces. The use of polarized light microscopy enabled examination of the layered structure, the distribution of pigments and binder in the paint system and also the pigment-binder interactions in cross-sections of the paintings.

## 2. Materials and Methods

### 2.1. Paint Materials and Oil Painting Mock-Ups

Five powdered mineral pigments of different colors and composition were selected to make a binary paint mixture with linseed oil as binder. The pigments were lead white (LW), orpiment (OR), cinnabar (CIN), azurite (AZ), and malachite (MAL). The characteristics of the pigments are shown in Table 1. All pigments were provided by Kremer Pigments GmbH and Co. KG (Aichstetten, Germany). According to the supplier, all of the pigments are single mineral pigments. OR, AZ and MAL were the coarsest pigments, with a particle size up to 120 μm, while LW and CIN were the finest pigments, with particle sizes less than 45 μm (Table 1). The organic binder used was Boiled Linseed Oil 026, supplied by Royal Talens (Apeldoorn, The Netherlands). The paint components were mixed in different proportions, as explained below, before being applied to a glass microscope slide (75 mm × 25 mm × 1 mm) with a paintbrush.

In order to determine the influence of the binder content on the reflectance, paint samples were produced using different pigment:binder ratios, and three mock-up paintings were prepared. After trial-and-error testing, we found that for all of the pigments, the optimal combination was 1 g of pigment and 1 mL of oil binder. Consequently, we made paint samples with less binder, by mixing 1 g of pigment and 0.5 mL of binder, and samples with excess binder, by mixing 1 g of pigment and 1.5 mL of binder. Two layers of paint were applied to glass microscope slides (75 mm × 25 mm × 1 mm) with a paintbrush, which provided an adequate level of opacity. The second layer was applied after one month, once the first layer was completely dry. A total of 15 painting mock-ups were prepared in this way. The mock-ups were identified by labels (written on the back of the glass slides)

indicating the pigment (LW, OR, CIN, AZ, MAL) followed by O (optimal pigment:binder combination), LS (low binder) or EX (excess binder).

**Table 1.** Properties (particle size and mineralogical composition) of the powdered pigments according to the supplier and determined in this study. The abbreviations of the names of the different pigment are also shown.

| Supplier Pigment Code | Abbreviated Pigment Name | Pigment Particle Size (μm), According to Supplier | Pigment Composition, According to Supplier | Pigment Particle Size (μm) * Determined in This Study | Pigment Composition [β] (wt.%) Determined in This Study | |
|---|---|---|---|---|---|---|
| Lead White 46000 | LW | <45 | Hydrocerussite $(PbCO_3 \cdot Pb(OH)_2)$ | 3  0.1–10 | Hydrocerussite $(PbCO_3 \cdot Pb(OH)_2)$ | 50 |
| | | | | | Cerussite $(PbCO_3)$ | 50 |
| Genuine orpiment K10700 | OR | <175 | Orpiment $(As_2S_3)$ | 84  1–270 | Orpiment $(As_2S_3)$ | 100 |
| Cinnabar, very fine 10624 | CIN | <20 | Cinnabar (HgS) | 12  0.4–40 | Cinnabar (HgS) | 100 |
| Azurite standard 10200.Deep greenish blue | AZ | <120 | Azurite $(Cu_3(CO_3)_2(OH)_2)$ | 22  0.2–55 | Azurite $(Cu_3(CO_3)_2(OH)_2)$ | 90 |
| | | | | | Quartz $(SiO_2)$ | <5 |
| | | | | | Malachite $(Cu_2(CO_3)(OH)_2)$ | <5 |
| Malachite K10300 | MAL | <120 | Malachite $(Cu_2(CO_3)(OH)_2)$ | 3 (70)  0.2–200 | Malachite $(Cu_2(CO_3)(OH)_2)$ | 95 |
| | | | | | Pseudomalachite $(Cu_5(PO_4)_2(OH)_4)$ | <5 |

*: primary maximum particle size and particle size range. Numbers in brackets represent the secondary maximum particle size. [β]: Mineralogy (wt.%) of pigments through semi-quantitative ($\pm$5%) XRD analysis.

In order to study the influence of the thickness of the paint layer on the reflectance, we used the optimal pigment: binder combination. Thus, 1 g of each pigment was mixed with 1 mL of linseed oil to make the diverse paint samples. Once the consistency of each mixture was deemed suitable, up to five layers were applied to the glass microscope slides (Figure 1). Each layer was applied when the previous layer was completely dry, i.e., after an interval of one month. The single layer mock-up paintings made with lead white (LW, Figure 1A) and azurite (AZ, Figure 1D) were translucent, so that the sample label was visible under the paints, mainly in the AZ sample. We have included these samples here for comparative purposes, i.e., to highlight differences between paints made with different pigments. In this case the painting mock-ups were identified by the pigment code (LW, OR, CIN, AZ, MAL) followed by the number of layers applied (1L to 5L). All painting mock-ups were prepared under laboratory conditions (20 $\pm$ 5 °C and RH 40 $\pm$ 10%).

*2.2. Analytical Approach*

In order to achieve the study objective (to determine the effects of the thickness of the layer and the binder proportion on the reflectance of the paint surface), the following approach was used:

The particle size and mineralogy of powdered pigments were studied before the painting mock-ups were prepared:

- Prior to the pigments being mixed with the binder, the particle size was determined with a laser particle size analyzer (Mastersizer 2000LF, Malvern Instruments, Malvern, UK). Pigments were dispersed in alcohol by very gently mixing in order to prevent agglomeration. For each pigment, the volume distribution of the grain size was

determined. Primary and secondary maximum particle sizes and the particle size range were determined following [14]. The presence of a secondary maximum particle size can significantly modify surface roughness and consequently alter surface physical properties such as reflectance [14].

- The pigment mineralogy was identified by X-ray diffraction (XRD) analysis (X'Pert PRO PANalytical B.V., Malvern, UK), with Cu-Kα radiation, Ni filter, 45 kV voltage, and 40 mA intensity. The exploration range was 3° to 60° 2θ, at a goniometer speed of 0.05° 2θ s$^1$. Each mineral was identified and semi-quantified (±5%) using XPowder software (Granada, Spain) [18].

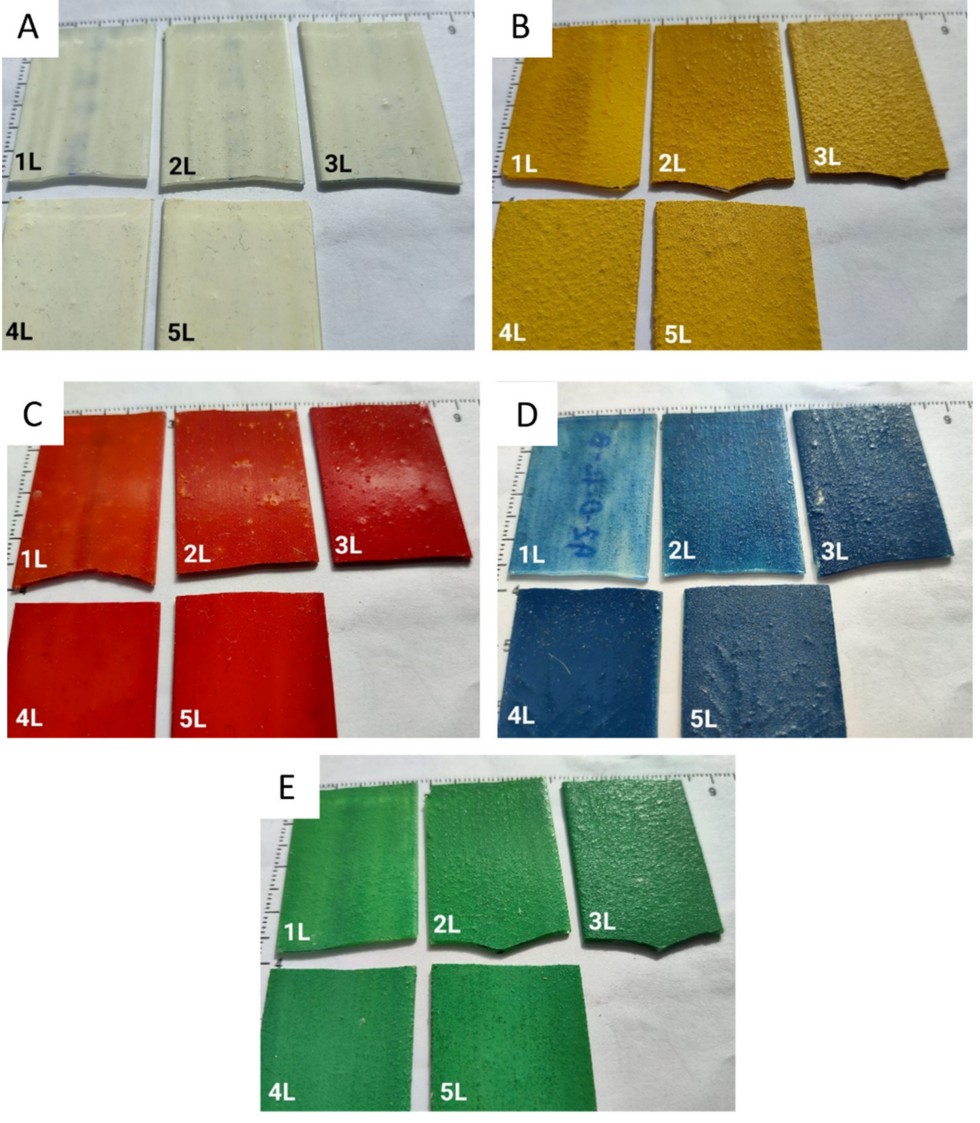

**Figure 1.** Painting mock-ups used to study the effect of the thickness of the paint layer (1L: one layer and 5L: five layers) on the reflectance. (**A**) Lead white (LW)-based oil painting mock-ups. (**B**) Orpiment (OR)-based oil painting mock-ups. (**C**) Cinnabar (CIN)-based oil painting mock-ups. (**D**) Azurite (AZ)-based soil painting mock-ups. (**E**) Malachite (MAL)-based oil painting mock-ups.

After characterization of the powdered pigments, the painting mock-ups were examined using imaging equipment consisting of a CCD sensor (Pulnix TM-1327 GE, PULNiX America, Inc., San José, CA, USA) (1040 rows, 1392 columns) with a 10 mm focal length objective lens, used in [14,19]. A spectrograph (ImSpector V10 Specim, Spectral Imaging Ltd., Oulu, Finland) with a spectral range of 400~1000 nm was fitted between the sensor and the lens. The spectral camera captured images of a linear array of 1392 pixels, with

the light spectrum at each pixel spread into 1040 bands. The resulting spectral resolution of camera and spectrograph was 4.55 nm. Optimal measuring conditions of the system were obtained in the spectral visible range 450–700 nm. Each mock-up was placed on a motorized 3D translation stage (Newport ILS-CC, Newport Corporation, Irvine, CA, USA), externally controlled by a motion controller (Newport MM4006, Newport Corporation, Irvine, CA, USA). The stage was able to be moved vertically. Therefore, the camera scanned the surface, line by line, in order to obtain an image at each of the 1040 bands. The light source was an incandescent lamp (Schott DCR® III, SCHOTT North America, Inc., NY, USA) with rectangular head of length 51 mm and width 0.89 mm. A cylindrical lens placed in front of the head focused the light. The illuminated area was similar to an ellipse with major axis of 15 cm and minor axis of 1 cm. Once the spectral images were obtained, the reflectance data were acquired in the MATLAB programming environment (The MathWorks, Inc., Natick, MA, USA). The pixel/mm ratio was estimated as a function of the distance between the sample and the camera, which, together with the vertical displacement of the object, allowed assignment of an image to each point on the surface. In order to obtain the reflectance spectra for each pixel, the spectral data were corrected to the dark noise (the room was kept dark) and normalized to a white Spectralon Wavelength Calibration Standard (Labsphere, Inc., North Sutton, NH, USA):

$$R(\lambda) = \frac{I_M(\lambda) - I_{BG}(\lambda)}{I_W(\lambda) - I_{BG}(\lambda)}$$

where $I$ is the intensity of each pixel of the mock-up ($I_M$), of the white reference ($I_W$) and background ($I_{BG}$).

In order to determine the possible changes in the shape of the reflectance spectra and in the reflectance value (intensity), the surfaces of the mock-ups were examined by stereomicroscopy, and cross-sections were examined by polarized light microscopy:

-　　Use of the stereomicroscope (SMZ 1000, Nikon, Istanbul, Turkey) allowed examination of the surface appearance of the painting mock-ups.
-　　Each mock-up was cut into two parts of different sizes with a diamond-tipped pen. The smallest part (25 mm × 25 mm × 1 mm) was embedded in resin (GTS Pro Soloplast) and cross-sectioned for transverse analysis of the paint layers. The thickness of the paint was measured in an Axioscope 5 (Zeiss, Jena, Germany) polarized light microscope.

## 3. Results and Discussion

### 3.1. Characterization of Powdered Pigments

Characterization of pigments (grain size and composition) should be carried out prior to determining the physical properties of painting surfaces or performing aging tests [14,20]. Therefore, we present the particle size and mineralogy of the pigments used here (Table 1), since these will determine the thickness of the paint layers (and thus the thickness of the painting), and the opacity/translucency of the painting (according to the refractive index of both the pigment and the linseed oil binder).

The particle sizes reported by the supplier were not as accurate as those determined by the laser particle size analyzer (Table 1), since the primary and also the secondary (when identified) maximum particle sizes, and the particle size range for each pigment should be included, as indicated in [14]. In the current research, the primary maximum particle size for all pigments was less than the maximum grain size reported by the supplier (Table 1). The coarsest (largest particle size) pigment was OR, followed by AZ and CIN, and the finest (smallest particle size) pigment was LW. Although the primary maximum particle size of the MAL pigment was 3 μm (as for LW), a secondary maximum particle size (ca. 70 μm) was also detected. The range of sizes detected for MAL pigment was relatively wide (0.2–200 μm).

Regarding the mineralogical composition, as reported in [14,20], the results of the XRD analysis showed that the information provided by the supplier was not accurate since impurities were detected in LW, AZ and MAL (Table 1). Thus, cerussite ($PbCO_3$) was identified in addition to hydrocerussite ($PbCO_3 \cdot Pb(OH)_2$) in LW; malachite ($Cu_2(CO_3)(OH)_2$) was detected in addition to azurite ($Cu_3(CO_3)_2(OH)_2$) in AZ, and pseudomalachite ($Cu_5(PO_4)_2(OH)_4$) was detected in MAL. Moreover, quartz ($SiO_2$) was detected in AZ, as previously reported [14]. The observed impurities indicate the natural origin of these pigments, since the compounds identified are not fillers (sometimes intentionally added to paint to reduce the costs) [20]. We cannot rule out the existence of other impurities, as if they are present at amounts of less than 5% wt., they would not be detected by XRD analysis.

The opaquest paints were those made with OR and CIN, followed by those made with MAL, LW and finally AZ, which produces the most translucent paints (Figure 1). The low hiding (covering) power is due to the proximity between the refractive index of the azurite pigment and the value for the linseed oil ($n_{azurite}$: 1.73–1.84 and $n_{linseed\ oil}$: 1.48, [21]). Lead white and malachite paints also exhibited low hiding power ($n_{malachite}$: 1.65–1.90; $n_{white\ lead}$: 1.94–2.09, [21]), while orpiment and cinnabar paints showed higher hiding power (opaquer paints) as a consequence of the refractive indexes ($n_{orpiment}$: 2.40–3.02 and $n_{cinnabar}$: 2.81–3.15), which were higher than for the linseed oil [21].

### 3.2. Influence of the Thickness of the Paint Layer on the Reflectance

The thickness of the paint layer increased with the number of applications (Table 2). The thickest layers corresponded to paints containing OR, AZ, and MAL while the layers of paints containing LW and CIN were generally thinner (the thicknesses of the LW- and CIN-based paints never exceeded 315 μm). The thickness of the paint layer is therefore related to the pigment grain size: the coarsest pigments produced the thickest layers. OR and AZ were the coarsest pigments, and MAL, in addition to a reduced primary maximum particle size of 3 μm, was characterized by a coarse secondary maximum particle size (70 μm) and a wide size range (0.2–200 μm).

The reflectance spectra of the samples with various paint layers and consequently of different thicknesses are shown in Figure 2. The reflectance measurement of the one-layer painting mock-up made with AZ-based paint is not representative, since this sample was translucent (AZ, Figure 1D), as indicated by the labelling being visible through the paint. Spectra for all of the mock-ups prepared with the same pigment did not reveal any difference in the reflectance, except for the AZ mock-ups with one layer and four layers (Figure 2D), as the former showed an intense reflectance band in the range 450–460 nm and the latter showed a large decrease in the intensity of bands assigned to the pigment over the whole wavelength range. In order to explain these trends in reflectance, the surfaces of the AZ samples were analyzed in detail (Figure 3) and the corresponding cross-sections were examined by polarized light microscopy (Figure 4). Due to the low hiding power of the azurite pigment, in the single layer AZ mock-up, the glass slide (substrate of the painting) was not completely covered by pigment grains (Figures 3P and 4P). As a result, the incident light on the mock-up surface was reflected by the glass slide, increasing the reflectance values. Moreover, the reflectance spectrum of this single layer AZ-based painting was slightly modified, as the glass influenced the reflectance measurements (Figure 2D). On the other hand, in the AZ mock-up with four layers, examination of the paint surface (Figure 3S) and of the cross-sections (Figure 4S) revealed accumulation of oil on the surfaces and a clear wavy pattern (Figure 3S). As result of the oil accumulation, the reflectance was greatly reduced due to the greater absorbance of the binder in the visible wavelength range relative to the pigments [10,13,17].

**Table 2.** Thickness of the paint layers (one to five layers, 1L to 5L), as measured by polarized light microscopy. LW, lead white-based oil painting mock-ups; OR, orpiment-based oil painting mock-ups; CIN, cinnabar-based oil painting mock-ups; AZ, azurite-based oil painting mock-ups; MAL, malachite-based oil painting mock-ups.

| ID | Thickness (μm) |
|---|---|
| LW-1L | $85 \pm 1$ |
| LW-2L | $110 \pm 1$ |
| LW-3L | $172 \pm 2$ |
| LW-4L | $214 \pm 1$ |
| LW-5L | $303 \pm 1$ |
| OR-1L | $125 \pm 1$ |
| OR-2L | $157 \pm 4$ |
| OR-3L | $188 \pm 6$ |
| OR-4L | $315 \pm 9$ |
| OR-5L | $504 \pm 2$ |
| CIN-1L | $59 \pm 1$ |
| CIN-2L | $78 \pm 2$ |
| CIN-3L | $163 \pm 3$ |
| CIN-4L | $238 \pm 3$ |
| CIN-5L | $315 \pm 5$ |
| AZ-1L | $81 \pm 5$ |
| AZ-2L | $172 \pm 7$ |
| AZ-3L | $347 \pm 9$ |
| AZ-4L | $441 \pm 5$ |
| AZ-5L | $441 \pm 8$ |
| MAL-1L | $116 \pm 4$ |
| MAL-2L | $188 \pm 5$ |
| MAL-3L | $219 \pm 5$ |
| MAL-4L | $315 \pm 3$ |
| MAL-5L | $347 \pm 3$ |

Two groups of samples were distinguished in the painting mock-ups according to the reflectance values (intensity):

1. For the LW mock-ups (Figure 2A), the reflectance increased over the whole wavelength range (450–700 nm), as the number of layers (and thickness of the overall coating, Table 2) increased.

2. For the other mock-ups (Figure 2B–E), no relationship was observed between the thickness of the coating and the reflectance value. Thus, for OR-based paint, the reflectance value was lowest for the mock-up with two layers ($157 \pm 4.5$ μm thick) and highest for the mock-up with three layers ($188 \pm 6.2$ μm thick) (Figure 2B). For the CIN-based paint, the reflectance value was lowest for the mock-up with 1 layer ($59 \pm 0.9$ μm thick) and highest for the mock-up with two layers ($78 \pm 1.9$ μm thick) (Figure 2C). For the AZ-based paint, the highest reflectance value (by far) corresponded to the mock-up with one layer ($81 \pm 5.5$ μm thick) due to the low hiding power as was explained before, and the lowest value corresponded to the mock-up with four layers ($441 \pm 4.6$ μm thick) (Figure 2D). For the MAL-based paint, although the differences in reflectance values were very small, the highest reflectance value corresponded to the mock-up with three layers ($219 \pm 5.1$ μm thick) and the lowest value corresponded to the mock-up with five layers ($347 \pm 2.9$ μm thick) (Figure 2E).

The difference between these two groups can be explained by the pigment-binder interaction and the paint texture, which in turns depends on the pigment grain size and the binder distribution. Examination of the LW mock-ups by polarized light microscopy revealed a homogenous mixture of pigment and binder (Figure 4A–E), which was also partially translucent, as observed in the samples with one and two layers (Figure 3A–B). In previous research, the transparency of the LW oil-based paint has been attributed to saponification of the oil in the presence of this pigment, leading to the formation of lead

linoleate in a hard and translucent paint film. Saponification can occur even 24 h after painting preparation [22]. This can be attributed to the subsequent partial disappearance of the white pigment particles [22–27] or to changes in the refractive index of the linseed oil due to aging processes [28]. However, it has been reported that a change in the refractive index of the oil binding medium is reduced and unlikely to influence the appearance of paintings, ruling out the idea proposed in [29]. As a result of the chemical changes caused by saponification, the light-scattering properties of the paint are reduced and light could deepen in the painting layer, creating a slight darkening [23–26] relative to the recently prepared painting. The darkening should explain the low reflectance as indicated in the Figure 2A for the WL-1 layer mock-up. However, the reflectance value increased as more layers were applied (increasing the thickness) (Figure 2A). This could be explained by the fact that the layers were added at intervals of 1 month (to allow the paint to dry), during which the paints may have lost their translucency (and their corresponding darkening), as observed in the micrographs (Figure 3B–E). Lighter colored paints have higher reflectance values. Furthermore, we cannot overlook the effect that the soaps formed in the drying process may have on the reflectance. Further studies should be performed to clarify this effect.

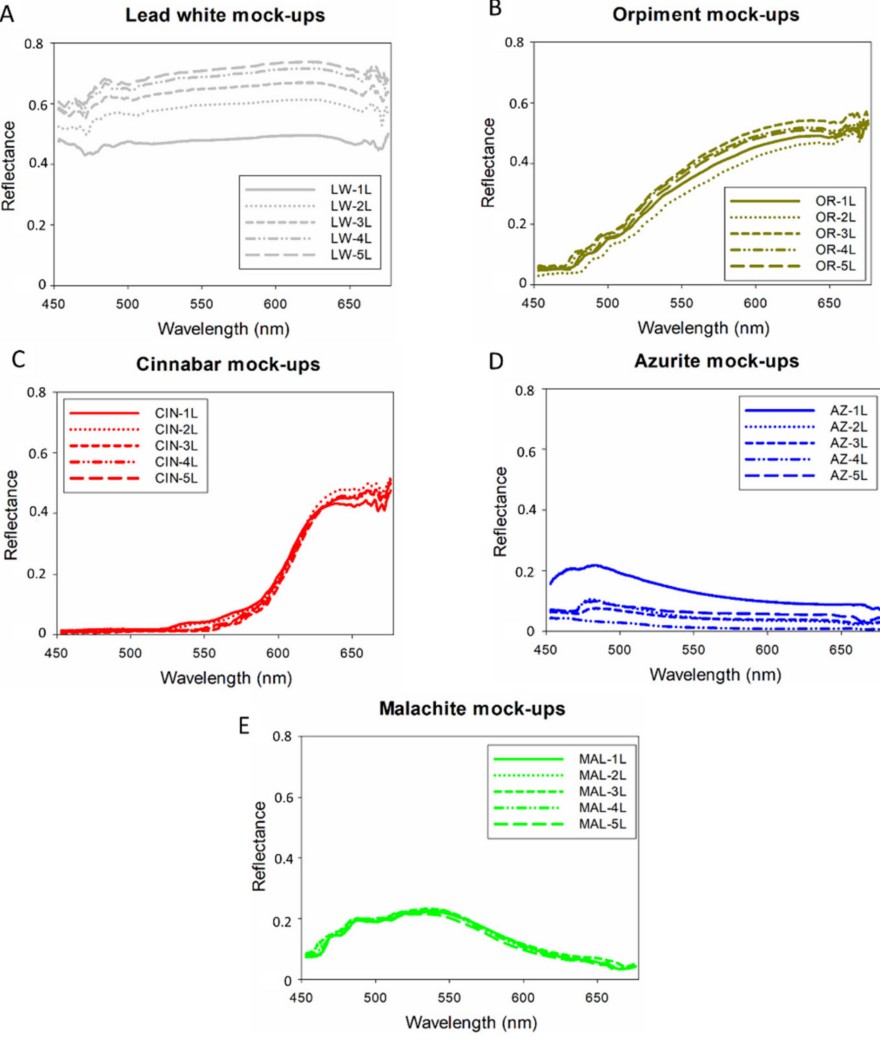

**Figure 2.** Reflectance spectra of the painting mock-ups used to determine the influence of the thickness of the paint layers (one to five layers, 1L to 5L) on the reflectance. (**A**) Lead white (LW)-based oil painting mock-ups. (**B**) Orpiment (OR)-based oil painting mock-ups. (**C**) Cinnabar (CIN)-based oil painting mock-ups. (**D**) Azurite (AZ)-based oil painting mock-ups. (**E**) Malachite (MAL)-based oil painting mock-ups.

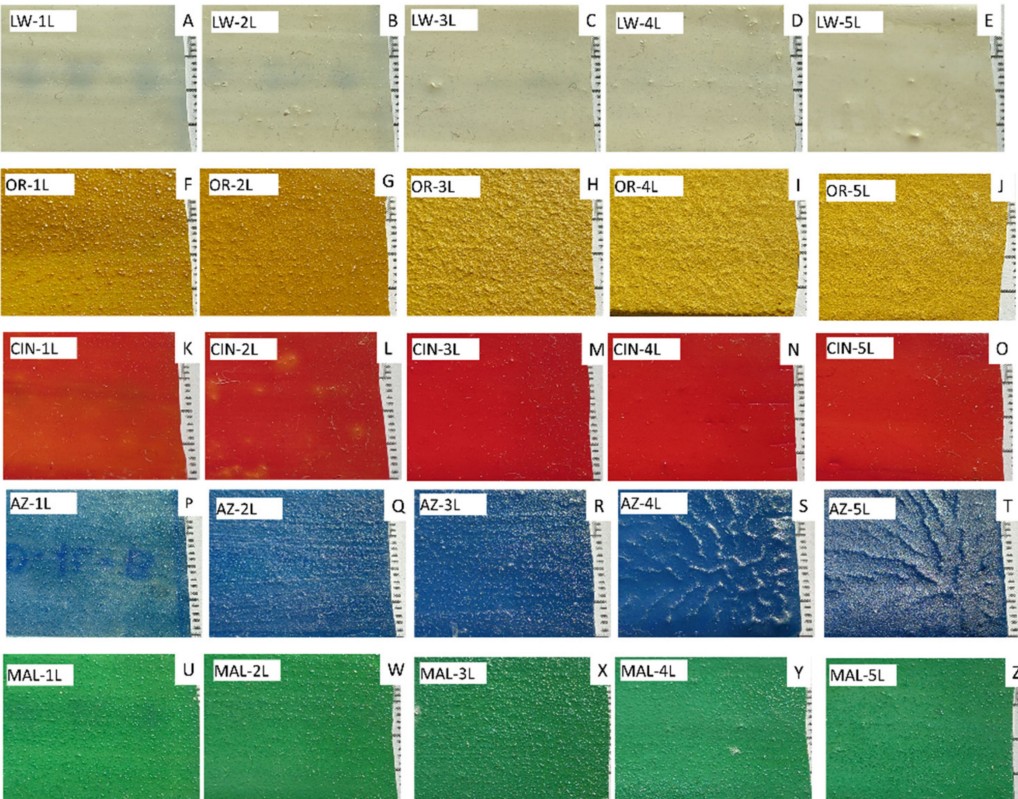

**Figure 3.** Digital photographs of the painting mock-ups used to determine the influence of the painting layer thickness (one layer to five layers, 1L to 5L) on the spectral reflectance. (**A–E**): lead white (LW)-based oil painting mock-ups. (**F–J**): orpiment (OR)-based oil painting mock-ups. (**K–O**): cinnabar (CIN)-based oil painting mock-ups. (**P–T**): azurite (AZ)-based oil painting mock-ups. (**U–Z**): malachite (MAL)-based oil painting mock-ups.

For the other painting mock-ups (OR, CIN, AZ and MAL), the lack of a direct relationship between the thickness of the paint and the reflectance values can be explained by the accumulation of oil on the surface of the paint preparation. It is already known that paints with greater accumulation of oil on the surface display lower reflectance due to the higher absorption levels [10,13,17]. The OR mock-ups with two layers yielded the lowest reflectance values, due to the darkening related to accumulation of oil on the surface (Figures 3G and 4G; note in Figure 4G arrows indicating accumulation of binder). However, in the OR sample with the highest reflectance value (OR-3L, Figures 3H and 4H), pigment grains were observed protruding from the binder in the external (superficial) layer (indicated by rectangles in Figure 4H); consequently, this surface (Figure 3H) was rougher than those of the other OR mock-ups. Pigments usually display higher reflectance values than binders [18]. In the OR mock-ups, the coarser grains appeared on top of the thin layers (corresponding to each application) composed by finer pigment particles (Figure 4F–J).

In the CIN mock-ups, the lowest reflectance was detected in the sample with 1 layer and was also related to the accumulation of oil on the surface (Figure 4K; note arrows indicating accumulation of binder). The highest reflectance, detected in the CIN mock-up with 2 layers, was associated with the presence of light orange areas on the surfaces (Figure 3L).

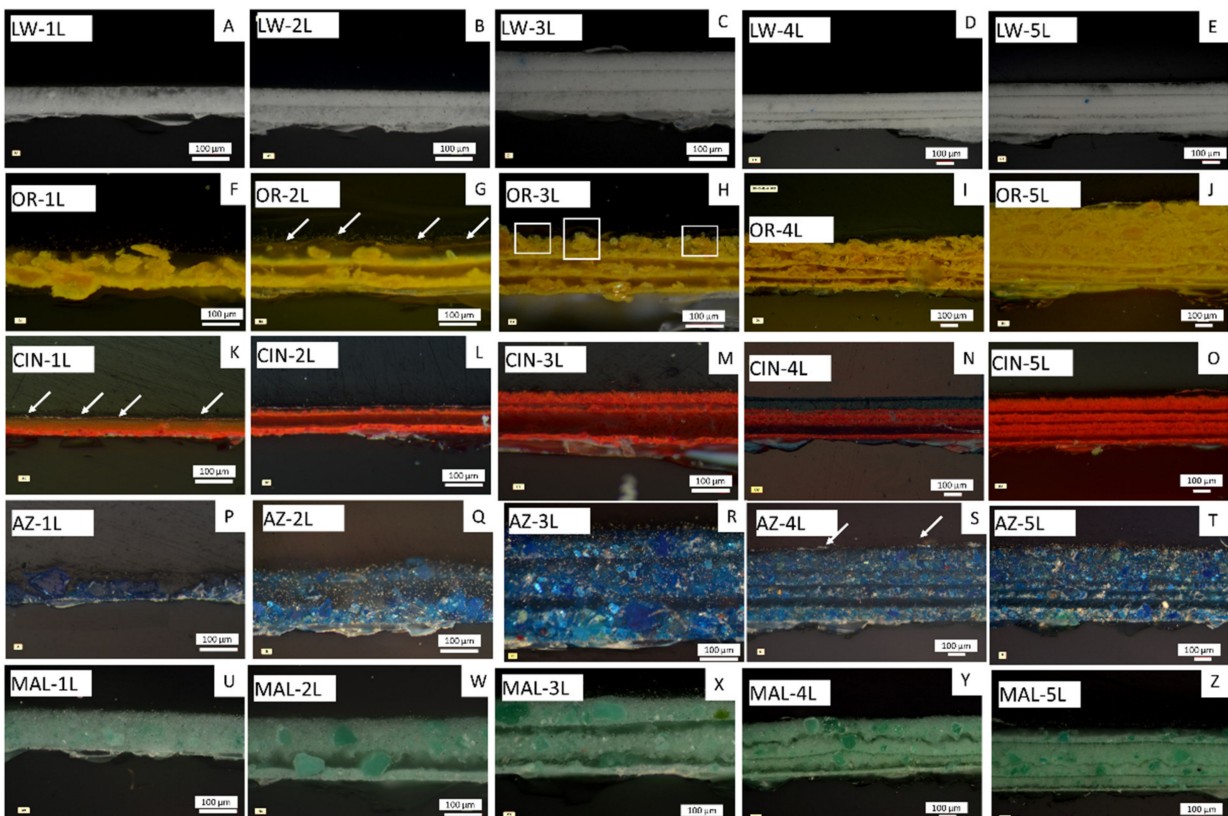

**Figure 4.** Cross-sections of the mock-ups under polarized light microscopy (transmitted light and cross polar modes). The samples were examined to determine the influence of the thickness of the paint layer (one layer to five layers, 1L to 5L) on the spectral reflectance. (**A–E**): lead white (LW)-based oil painting mock-ups. (**F–J**): orpiment (OR)-based oil painting mock-ups. (**K–O**): cinnabar (CIN)-based oil painting mock-ups. (**P–T**): azurite (AZ)-based oil painting mock-ups. (**U–Z**): malachite (MAL)-based oil painting mock-ups. Arrows indicate the accumulation of binder and rectangles indicate grains protruding from the binder.

The AZ mock-up with 4 layers showed the lowest reflectance due to the darkening and the wavy texture (Figure 3S) caused by accumulation of oil on the surface, as detected in the cross-sections (Figure 4S; note arrows indicating accumulation of binder). The AZ mock-up with 1 layer displayed the highest reflectance (as previously mentioned).

The MAL mock-ups with three layers (Figure 3X) was brighter than the other MAL mock-ups, leading to detection of slightly greater reflectance in this sample. Examination of cross-sections of the MAL mock-ups revealed a mixture of larger crystals surrounded by a matrix consisting of much smaller grains (Figure 4U–Z). Although accumulation of oil was observed between layers, accumulation of binder was not observed in the cross sections of the external layer (Figure 4U–Z).

Moreover, the OR, CIN, and MAL mock-ups yielded reflectance spectra which were more similar to each other than to those yielded by the LW and AZ mock-ups, in which the influence of the pigment-binder interaction (LW pigment) and hiding power of the paint (AZ pigment) were found to be important. As the oil accumulation depends on how the paint is applied (i.e., it depends on the painter's skill), the reflectance spectra are expected to vary.

*3.3. Influence of the Oil Quantity on the Reflectance*

Regarding the effect of the oil quantity on the reflectance of each surface (Figure 5), with the exception of the AZ mock-ups, the reflectance values and shapes of the reflectance spectra were similar in all paints prepared with the same pigment. In the case of AZ

mock-ups, the sample with a deficit of binder (AZ-LS) showed a shift in the spectrum in the first range of the visible wavelengths (around 470 nm, Figure 5D), which could be due to the greater reflection of the light from the substrate as the amount of binder appeared to be insufficient to cause agglomeration of the azurite grains, and consequently a greater amount of light reached the substrate. The AZ mock-up with excess binder showed a large decrease in reflectance (Figure 5D) due to accumulation of the binder on the surface, resulting in a wavy texture of the painting (Figures 6L and 7L; arrows indicate the accumulation of binder).

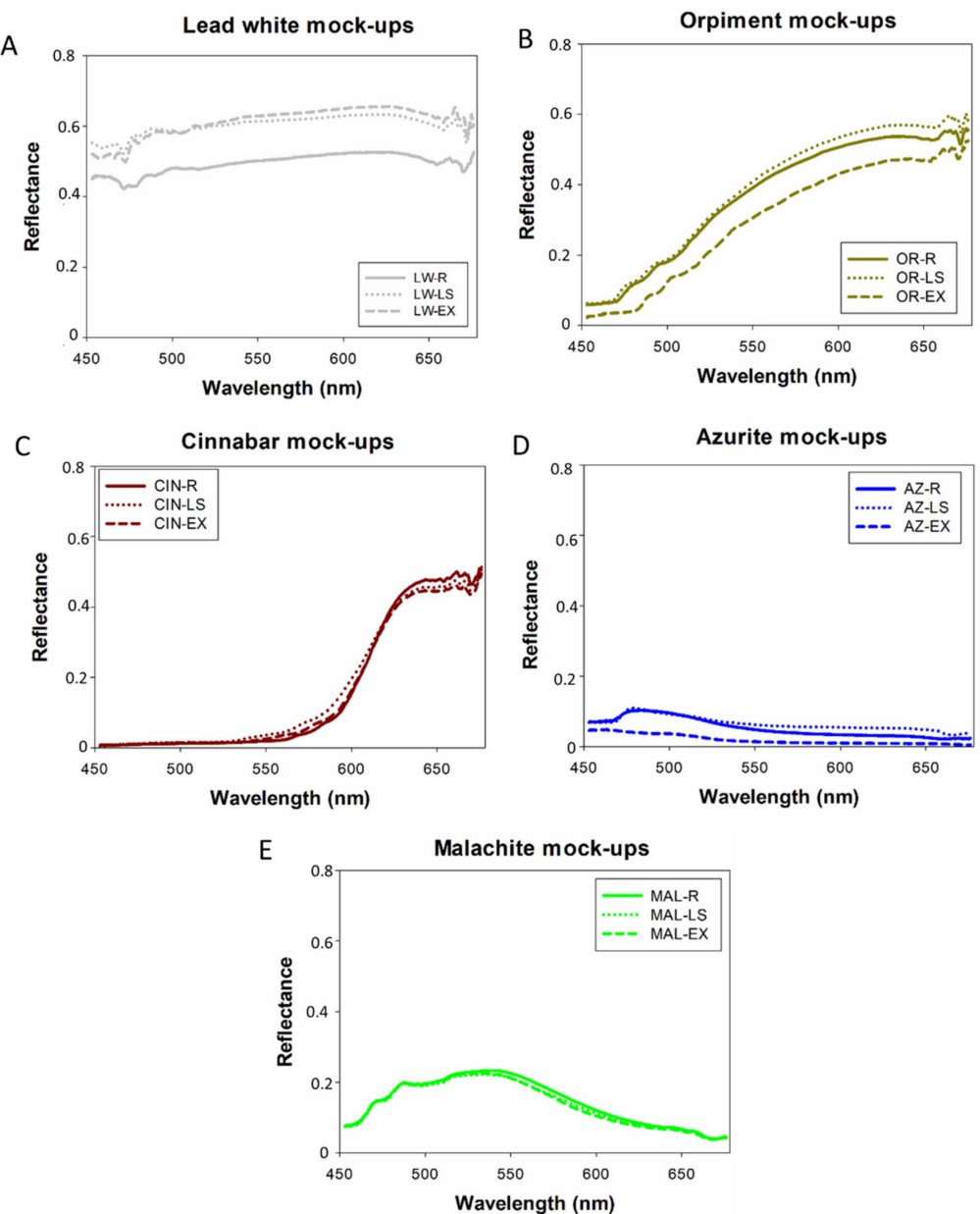

**Figure 5.** Reflectance spectra of the painting mock-ups used to determine the influence of the binder quantity (O-optimal quantity-, LS-low binder- and EX-excess binder-) on the reflectance. (**A**) Lead white (LW)-based oil painting mock-ups. (**B**) Orpiment (OR)-based oil painting mock-ups. (**C**) Cinnabar (CIN)-based oil painting mock-ups. (**D**) Azurite (AZ)-based oil painting mock-ups. (**E**) Malachite (MAL)-based oil painting mock-ups.

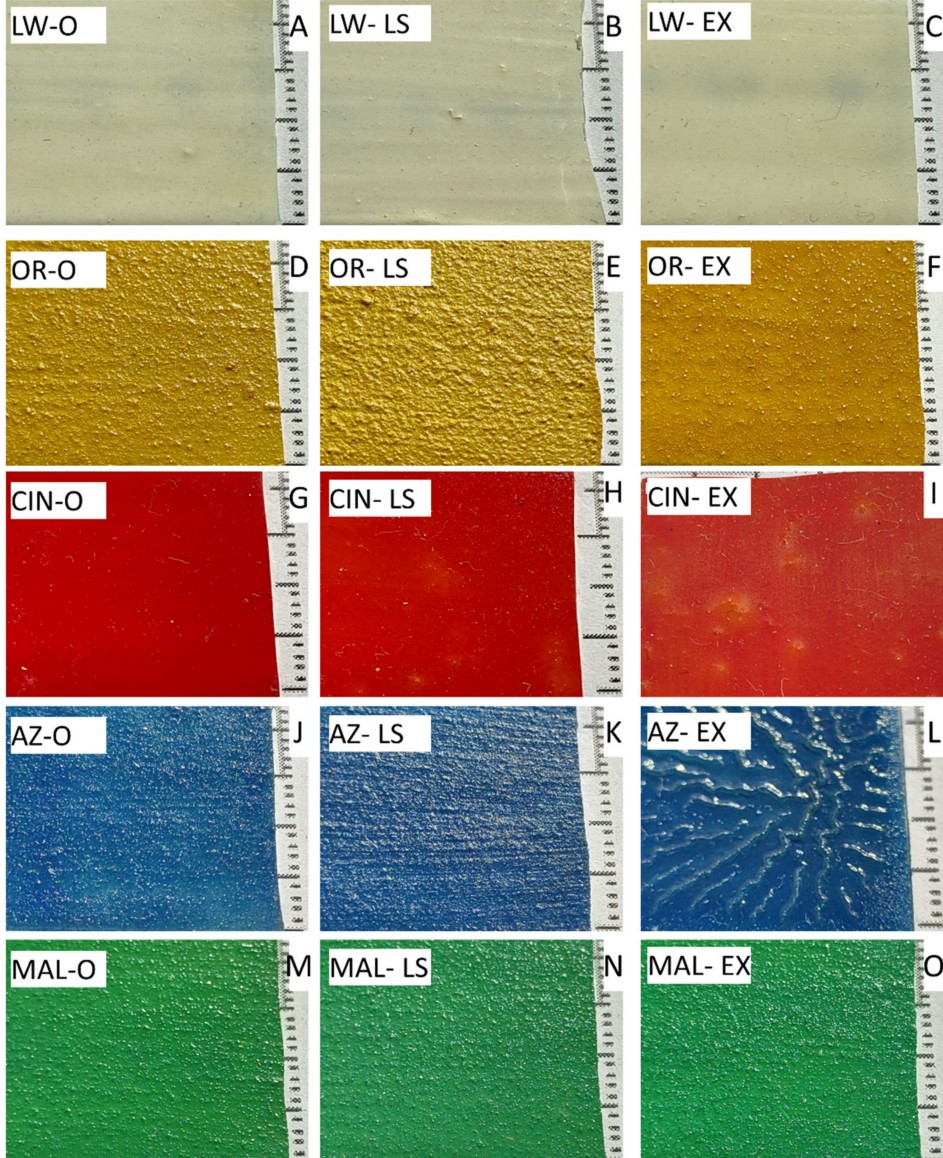

**Figure 6.** Digital photographs of the painting mock-ups used to determine the influence of the binder quantity (O-optimal quantity-, LS-low binder- and EX-excess binder-) on the reflectance. (**A–C**): lead white (LW)-based oil painting mock-ups. (**D–F**): orpiment (OR)-based oil painting mock-ups. (**G–I**): cinnabar (CIN)-based oil painting mock-ups. (**J–L**): azurite (AZ)-based oil painting mock-ups. (**M–O**): malachite (MAL)-based oil painting mock-ups.

Considering the reflectance value, for the OR and AZ mock-ups, samples with excess binder showed the lowest reflectance, and the samples with a deficit of binder showed the highest reflectance (Figure 5B,D) as expected. This is due to the fact that binders have higher absorbance levels than pigments [10,13,15]. The surfaces of the painting mock-ups with excess binder were darker (Figure 6F,L) due to the accumulation of oil (Figure 7F,L). However, this trend was not detected for the LW, CIN and MAL samples (Figure 5A,C,E). In the LW-mock-ups, the paint with the optimal amount of binder (LW-O) showed the lowest reflectance and the sample with excess binder (LW-EX) showed the highest reflectance value (Figure 5A). Analysis of the cross-sections of the painting mock-ups revealed a continuous, homogenous layer of a well-mixed pigments on the glass slide holder in the LW mock-ups (Figure 7A–C). Although less intense than in the paints containing other pigments, a slight accumulation of binder was detected in the uppermost part of the LW-based paint (Figure 7C, arrows showing the accumulation of binder). Moreover, formation

of soap due to reaction between the linseed oil and the LW pigment [22–27] should be more intense in the sample with highest amount of binder, and these soaps should be considered as affecting the reflectance values. Soap formation occurred even at 24 h after mock-up preparation [22].

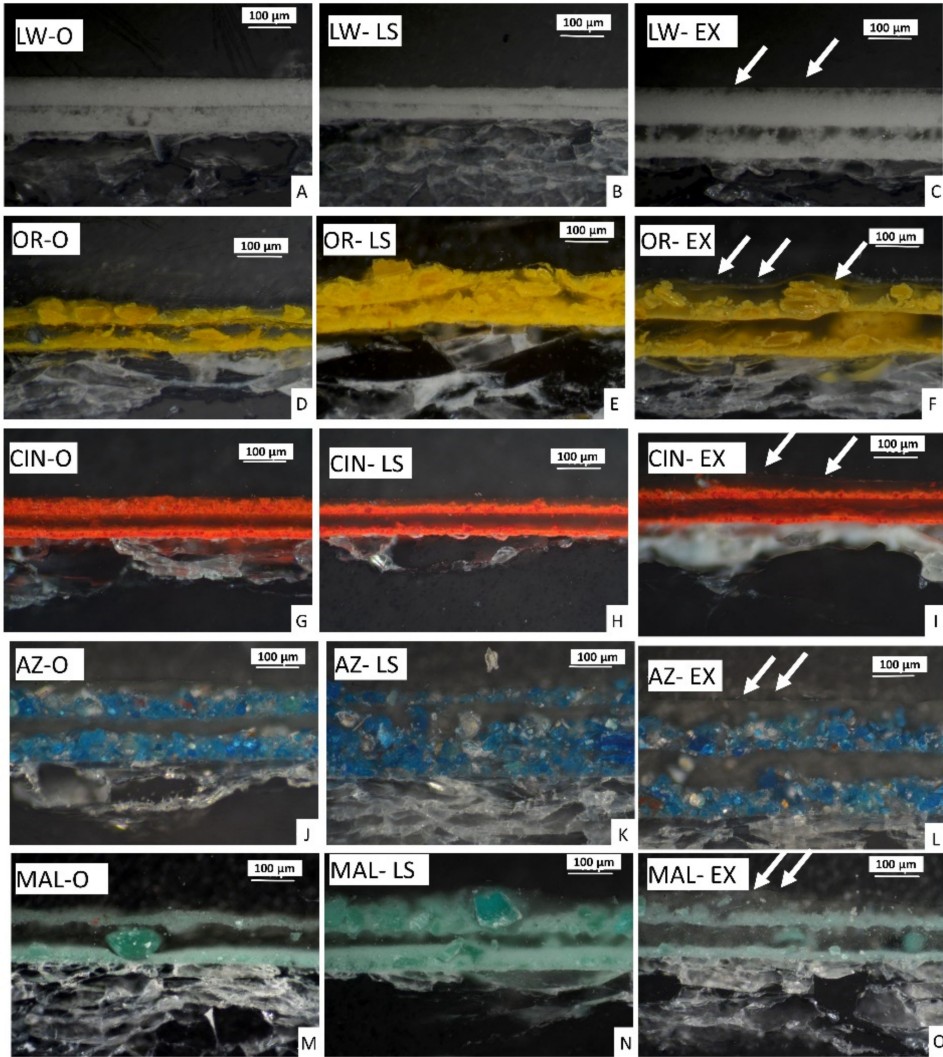

**Figure 7.** Cross-sections of mock-ups under polarized light microscopy. The samples were examined to determine the influence of the binder quantity (O-optimal quantity-, LS-low binder- and EX-excess binder-) on the reflectance. (**A–C**): lead white (LW)-based oil painting mock-ups. (**D–F**): orpiment (OR)-based oil painting mock-ups. (**G–I**): cinnabar (CIN)-based oil painting mock-ups. (**J–L**): azurite (AZ)-based oil painting mock-ups. (**M–O**): malachite (MAL)-based oil painting mock-ups. Arrows indicate the accumulation of binder.

For the CIN and MAL mock-ups, the spectra of the samples with excess binder showed the lowest reflectance value, and those with the optimal amount displayed the highest reflectance value (Figure 5C,E). Examination of the samples by stereomicroscopy and polarized light microscopy enabled identification of mock-ups with excess binder or accumulation of binder on the surface (Figure 6I,O and Figure 7I,O; arrows showing the binder accumulation), which caused a reduction in the reflectance [13].

For the CIN and MAL mock-ups (Figure 5C,E), the spectra of the paintings with different binder amounts were more similar to each other than to the spectra of the other paint mock-ups. Moreover, the spectra overlapped throughout the wavelength range. However, as the amount of oil that accumulates on the painting surface depends on

how it is applied (as mentioned in the previous section), different reflectance spectra can be expected.

## 4. Conclusions

Previous research has shown that the reflectance of paintings, as determined by hyperspectral imaging, is affected by diverse parameters such as pigment impurities and grain size. The present study did not exhaustively evaluate all of the parameters that affect the reflectance, but focused specifically on the influence of the thickness of the paint layer and the binder quantity, in terms of the shape of the reflectance spectrum and the intensity of the reflectance. The painting mock-ups tested consisted of paints made with mixtures of a single pigment (lead white, orpiment, cinnabar, azurite or malachite) and linseed oil (binder) applied to glass microscope slides.

The findings revealed that the thickness of the paint layers and the binder proportion in the paint system did not directly influence the reflectance, as it was not possible to identify a direct relationship between thickness or amount of binder and the reflectance of the painted surfaces. However, the reflectance (mainly the value) depended on the pigment-binder interaction and accumulation of binder on the surface of the painting mock-ups, regardless of the amount of binder used in the paint samples. Regarding the effect of the pigment-binder interaction, the metal soaps due to a reaction between the linseed oil and the lead white pigment, increased the translucency and darkness of the paint, leading to a lower reflectance than that of the recently painted surface. However, the application of additional layers to make thicker paint samples increased the reflectance, since the paints became less translucent and darker. Moreover, the changes in reflectance can be explained by the hiding power of the diverse pigments tested. Also, in samples in which greater amounts of binder accumulated on the surface (not necessarily those paints containing the highest amounts of binder), the reflectance decreased due to greater absorption by the linseed oil than by the pigment present in the mixture.

All in all, here we demonstrate that when spectral data are acquired using imaging techniques, several characteristics of the paint components (pigment and binder) and of the paint system itself (including pigment-binder interactions) should be taken into account to enable correct interpretation of the results. Consequently, application of complementary analytical techniques (as conducted here) is strongly recommended. Non-invasive hyperspectral imaging is highly applicable to the conservation of cultural heritage, when the effects of the different parameters on the reflectance spectra are known. The present findings contribute novel information regarding these effects.

**Author Contributions:** Conceptualization, J.S.P.-A. and C.C.; methodology, J.S.P.-A. and C.C.; software, J.S.P.-A. and C.C.; validation, J.S.P.-A. and C.C.; formal analysis, J.S.P.-A., C.C., S.S. and J.M.R.; investigation, J.S.P.-A., C.C., S.S. and J.M.R.; resources, J.S.P.-A. and C.C.; data curation, J.S.P.-A., C.C., S.S. and J.M.R.; writing—original draft preparation, J.S.P.-A.; writing—review and editing, J.S.P.-A., C.C., S.S. and J.M.R.; visualization, J.S.P.-A. and C.C.; supervision, J.S.P.-A. and C.C.; project administration, J.S.P.-A. and C.C.; funding acquisition, J.S.P.-A. and C.C. All authors have read and agreed to the published version of the manuscript.

**Funding:** This research was funded by the University of Vigo, through the CINTECX Research Center, and by the University of Granada, through the Research Group of the Junta de Andalucía RMN-179. J.S. Pozo-Antonio was supported by the Ministry of Science and Innovation, Government of Spain through grant number RYC2020-028902-I.

**Institutional Review Board Statement:** Not applicable.

**Data Availability Statement:** Not applicable.

**Acknowledgments:** The authors thank Alberto Ramil from the University of A Coruña for technical assistance.

**Conflicts of Interest:** The authors declare no conflict of interest.

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
