# Peer review of "Reflectance of Oil Paintings: Influence of Paint Layer Thickness and Binder Amount"

_coatings, doi:10.3390/coatings12050601_

Round 1

Reviewer 1 Report

This paper evaluates the influence of the painting layer thickness and the binder concentration on the reflectance (spectral features and intensity) in painting mock-ups made with mixtures of pigments (lead white, orpiment, cinnabar, azurite or malachite) and linseed oil (binder). Paint layers were applied on glass microscope slides with superimposed brushstrokes to produce different thicknesses.

Although the objective of the research is comprehensible, the achievements reported in the conclusions are not exhaustive and are based on a questionable interpretation of the results. Moreover, in my opinion, the method used for samples preparation and the explanation of the spectral behaviour is not convincing and should be more clearly described or – better – changed.

A more precise and reproducible procedure for mock-ups preparation is crucial to provide reliable results.   

My main remarks are listed below:

Methods:

line 128: I am not sure that the method used here (successive superimposed brushstrokes) is the most reliable to obtain different thickness, especially because the drying process of the oil binder is time-consuming and the superposition of fresh layers can be hardly controlled.

Thickness measurements with stereomicrograph: Fig. 4 is very confusing.

Check all reference numbering: references 1-8 are from the journal template.

Line 187-88: “the spectral data were corrected to the dark noise”: how was the dark measured?

Equation 1: in standard reflectance spectroscopy measurements, the  equation including the reflectance ρ is generally used.

Line: 196: how can you study the color of the painting with the stereomicrograph? No colorimetric measurements are reported.

Line 197: I am not convinced of the thickness measurement procedure. Generally, the drying of oil-paint takes more than a month – especially if the relative quantity of the binder exceeds that of the pigment. How could you extract samples of approximately 4 mm × 26 mm × 1 mm from the fresh layers without deforming them?  

Results:

Line 208: “In the current research, the primary maximum particle size for all pigments was less than the maximum grain size reported by the supplier”: is it possible that the grain size declared by the manufacturer refers to the pigment in powder, i.e., it was measured before the pigment was dispersed in the binder? The preparation of the mixture may imply a reduction of the particles size and this would explain why the maximum particle size measured here is lower than the one declared.

Line 228-233: Are the reported refractive index values found in literature or did you measure them in this study? If they are from literature, please add some references.

Line 251: “Due to this transparency, the incident light on the surface was reflected by the glass slide, thus inducing an increase in reflectance”: I have some doubts about this. If the glass slide reflected the light, this effect would be observed also for the ZW 1 layer sample. Would it be possible that the higher reflectance is due to the material where the coverslip was laid for the measurement, and not to the glass? Besides, for AZ, the spectrum of the 1-layer sample shows features that are consistent with azurite in oil binder, while the spectra of the thicker layers do not show those similarities. Can you explain this? In lines 355-56, you reports that “the AZ mock-up with excess binder showed an important decrease in reflectance (Figure 5D) due to accumulation of the binder, accompanied by a wavy texture”: could the accumulation of the binder be in some way related also to the different shape of the spectra?

Line 380: “However, as was reported in the previous section, since the oil accumulation on the paint is dependent on the worker´s performance, other reflectance spectra distribution would be found.” I don’t understand this sentence, can you clarify it?

Minor corrections:

Line 58: “In addition, the refractive index of a pigment particle is not necessarily the same in all directions because light does not travel at the same speed in all directions”: this statement need to be rephrased otherwise it sounds not correct.

line 62: light scattered

line 63: “the greater the light scattering, the greater the paint”: something is missing here

line 79: Do you mean linseed oil emulsified with Arabic gum? Otherwise, Arabic gum is commonly used as a binder for tempera paints.

Line 83: the reflectance value, maybe.

Line 108: … “evaluate the effects of painting layer thickness and binder concentration”, please specify on what.

Line 196: “of” is missing.

Reviewer 2 Report

The authors are presenting two research questions (the thickness and the binder concentration) and they use nowadays techniques to try to answer them. 

While the techniques seem to be mastered at a professional level, especially the research question regarding the binder concentration seems somehow onesided: a paint is a system, actually a solution, where both binder and pigment are equally important. I am wondering how one can focus only on the concentration of the binder without considering that once the quantity of the binder modifies, the concentration of the pigment changes as well. 

I would be curious to read the take of the authors regarding this comment.

Generally speaking, the paper has a flowing style, but now and then the reader might difficulties getting what the authors meant.

For instance, the introduction part is difficult to follow. The explanation are too detailed for the scope the scope of the paper (the introduction should not be a crash course in optics) and now and then there are errors: 

  • paragraph 29 states that "incident light reflects in the same direction" - this is only the case when the incident light falls perpendicularly on the surface, which is a particular case.
  • paragraph 36 states that "the reason of the color is the separation of light beam" - this is incorrect since the color appears due to the selective absorption of the light,
  • paragraph 39 states that "the refraction index depends on the angles at which the light is refracted" - this is incorrect since it is the index dictating the angle 
  • in paragraph 54, the authors state " the light will be absorbed...suffering refraction" - absorption and refraction are two different phenomena,
  • in paragraphs 70, the authors mention "degradation forms" - not sure what is the meaning of it,
  • the paragraphs 77-90 are very difficult to grasp: it seems strange the speak of the intensity of the whole spectrum, but it might be a matter of the writing style; 

In the results part, there are several issues:

  • the photographs from Figure 3 do no present the feature the authors mention in the text (like the name of the sample marked on the slide)
  • same for figure 4
  • the features authors refer to regarding figure 4 can't spotted: the authors show indicate them by arrows,
  • authors mention several places "the accumulation of biding media on the surface",  but it is difficult to understand how did they evaluate this happening,
  • the author mention several places the saponification process: this is the solid state reaction happening in time and here the mock-ups were analysed after one month, so it is highly doubtable the saponification occurred to a such degree that it impacts on the paint quality. Or, if it did, the author have to present a proof.
  • how many replicates did the author prepare?
  • the features mentioned in paragraph 305 are not visible in the photographs
  • paragraph 302 mentions "grains ..protruding...in the external layer" - not clear what external layer is,
  • the features mentioned in paragraph 318 are not visible,
  • in paragraphs 351-356, the authors mention a shift in spectrum, but do not explain why (glass slide reflection?);  as well, they do assume the decrease in reflectance is due to accumulation of the binder, but they do not discuss the contribution of the glass slide which created issues for the same pigment (the thickness experiment),
  • in paragraphs 366-369, authors state the binder accumulation may be related to saponification: soaps and binder are two different compounds, so there is little logic in the statement.

In conclusions, the authors speak about the pigment mineralogy, but they haven't discussed anywhere about the influence of the impurities - either they do, they the drop mentioning it.

  • as well, it is not clear (paragraph 412) how the saponification influences reflectance: why an increase in transparency would increase the darkness? 
  • the meaning of the last paragraph (423-427) escapes me. 

The list of references is poorly written. It is unprofessional to refer several places in the text to missing bibliography: references 1-8.

Round 2

Reviewer 1 Report

The authors made several changes which improved the quality of the manuscript and solved the major issues. Although the novelty and scientific soundness of the manuscript are still quite low, I think that this research can add at some extend useful information to what has been already published on this topic.